# Nimodipine Used with Vincristine: Protects Schwann Cells and Neuronal Cells from Vincristine-Induced Cell Death but Increases Tumor Cell Susceptibility

**DOI:** 10.3390/ijms251910389

**Published:** 2024-09-27

**Authors:** Maximilian Scheer, Mateusz Polak, Saskia Fritzsche, Christian Strauss, Christian Scheller, Sandra Leisz

**Affiliations:** Department of Neurosurgery, Medical Faculty, Martin Luther University Halle-Wittenberg, Ernst-Grube-Str. 40, 06120 Halle (Saale), Germanysaskia.fritzsche@uk-halle.de (S.F.);

**Keywords:** nimodipine, neuropathy, vincristine, neuroprotection, Schwann cells, neuronal cells, non-small lung cancer, squamous tongue cancer, glioblastoma

## Abstract

The chemotherapeutic agent vincristine is commonly used for a variety of hematologic cancers, as well as solid tumors of the head and neck, bronchial carcinoma, as part of the procarbazine, lomustine and vincristine (PCV) regimen, for glioma. Damage to nerve tissue (neuropathy) is often dose-limiting and restricts treatment. Nimodipine is a calcium antagonist that has also shown neuroprotective properties in preliminary studies. In this approach here, we investigated the effects of the combination of vincristine and nimodipine on three cancer cell lines (A549, SAS and LN229) and neuronal cells (RN33B, SW10). Fluorescence microscopy, lactate dehydrogenase (LDH) assays and Western blot analyses were used. Nimodipine was able to enhance the cell death effects of vincristine in all tumor cells, while neuronal cells were protected and showed less cell death. There was an opposite change in the protein levels of Ak strain transforming/protein kinase B (AKT) in tumor cells (down) and neuronal cells (up), with simultaneous increased protein levels of cyclic adenosine monophosphate response element-binding protein (CREB) in all cell lines. In the future, this approach may improve tumor response to chemotherapy and reduce unwanted side effects such as neuropathy.

## 1. Introduction

Vincristine (VCR) is a common chemotherapeutic agent of the vinca alkaloid class, which prevents the assembling of microtubules in the mitotic spindle [1]. This causes mitosis disruption and cell death at metaphase [2]. It is used in a variety of oncologic conditions, including head and neck tumors [3,4,5], lung cancer [6,7,8], or gliomas as part of PCV (procarbazine, chloroethyl cyclohexyl nitrosourea (lomustine) and vincristine) regimes [9,10,11].

However, the activity of VCR is not specific or limited to tumor cells. As a result, side effects such as polyneuropathy or cranial nerve deficits and even optic atrophy may occur due to damage to cells of the nervous system [12,13,14,15,16]. VCR-induced peripheral neuropathy affects up to 80% of patients and is the main dose-limiting toxicity [16,17]. A substance used for neuroprotection in combination with VCR has not yet been established. To date, there are only a few preclinical studies that attempt to reduce the neurotoxicity of VCR [18,19].

Another problem with the majority of chemotherapeutic agents is the development of resistance [6]. This can be caused by various factors, such as efflux pumps, reduced activation of the prodrugs or tumor stem cells [20,21]. In this case, several chemotherapeutic agents are usually combined [22].

One compound that has shown neuroprotective properties in other settings is nimodipine (NIM). NIM is a calcium antagonist which was originally developed as an antihypertensive agent [23]. Due to the relaxing effect on the smooth musculature of the vessels, it is used for prophylaxis and treatment of vasospasm in patients with subarachnoid hemorrhage (SAH) [24]. Previous works of our group have demonstrated the neuroprotective effects of NIM on neuronal cells, astrocytes or Schwann cells under different conditions [25,26,27].

In preclinical experiments, our research group was able to show that NIM protects neuronal Schwann cells and astrocytes from oxidative, osmotic and heat stress. This effect correlated with an increased expression of protein kinase B (AKT) and cyclic adenosine monophosphate response element-binding protein (CREB) [25]. Similar effects were seen when electrospun NIM-loaded fibers were applied to these cell lines [27]. In the meantime, NIM is also used in our clinical routine: During microsurgery of vestibular schwannomas, NIM improves hearing outcomes [26]. In addition, we were able to show in another study that NIM is able to protect auditory hair cells from cisplatin-induced cell death, which was also associated with increased expression of AKT and CREB [28].

In order to investigate whether NIM also has neuroprotective properties in neuronal cells during chemotherapy stress with VCR and what effects of this combination with tumor cells can be observed, we conducted this study. For this purpose, we used different tumor cell lines as well as neuronal cells and Schwann cells and treated them with VCR with or without pretreatment with NIM.

## 2. Results

### 2.1. NIM Treatment Affects Cell Morphology and Cell Viability during VCR Treatment

At first, we investigated the influence of VCR treatment and the combination of VCR with NIM on cell morphology. Cells were cultured in the absence (control) or presence of 1 VCR for 24 h. For the combination with NIM, pretreatment with 20 NIM for 24 h was performed. The control was treated with the same amount of ethanol. In addition, staining with DAPI for visualization of cell nuclei and CellROX™ for visualization of oxidative stress was performed.

The benign cell lines RN33B (Figure 1a) and SW10 (Figure 1b) showed a high confluence in the controls and a significant reduction in these after treatment with VCR. Furthermore, the morphology was altered. The cells appeared more granular and sporadic. In combination with NIM, there was an increase in confluence compared to treatment with VCR alone. Abnormally shaped cells continued to appear but were more normally configured than without NIM treatment. Treatment with VCR led to a decrease in the amount of DNA (NucBlue™). By combining it with NIM, we saw a reduction in these effects. The staining with CellROX™ showed a decrease under treatment with VIN and a re-increase in the combination treatment, analogous to nuclei staining. The data could be confirmed via quantification (Appendix A).

Malignant cell lines A549 (Figure 2a), SAS (Figure 2b) and LN229 (Figure 2c) each showed high confluence and regular cell morphology in controls. After treatment with VCR, the confluence was reduced. There were markedly more apoptotic cells. The combination treatment with NIM led to a further reduction in confluence. More abnormally shaped apoptotic cells were visualized than with treatment with VCR alone. In the malignant cells, VCR treatment also led to a decrease in DNA quantity (NucBlue^™^). However, we saw an increase in these effects when combined with NIM. The staining with CellROX^™^ showed a decrease under treatment with VIN and a further decrease in the combination treatment, analogous to nuclei staining. The data could be confirmed via quantification (Appendix A).

### 2.2. NIM Protected Neuronal Cells and Schwann Cells during VCR Treatment

For the calculation of cytotoxicity, lactate dehydrogenase (LDH) assays were used in which the absorption of the culture medium (background control) was subtracted from the values. To normalize the data, the untreated cells were lysed with Triton X100 and the absorbance of the total lysis was set to 100%.

In general, we observed opposite effects in the benign and malignant cell lines when VCR was combined with NIM. The benign cell lines RN33B and SW10 showed no cell death in the control and after NIM treatment alone. In the RN33B, cell death rate was 3.65 ± 0.18% after 24 h and 12.60 ± 4.63% after 48 h in the controls. The NIM-treated samples showed comparable values with a cell death rate of 3.89 ± 0.19% after 24 h and 12.45 ± 4.56% after 48 h (Figure 3a). The SW10 cells showed a cell death rate of 1.89 ± 0.14% after 24 h and 11.53 ± 2.01% after 48 h in the controls. The NIM-treated samples showed similar values with a cell death rate of 4.09 ± 0.36% after 24 h and 10.78 ± 0.94% after 48 h (Figure 3b). As expected, the addition of VCR showed a significant increase in cell death and, interestingly, a decrease in cell death when combined with NIM. In detail, RN33B cells treated with VCR showed a cell death rate of 23.33 ± 1.68% after 24 h and 70.56 ± 9.72% after 48 h, respectively. The combination with NIM reduced cell death to a rate of 12.23 ± 0.17% after 24 h and 50.25 ± 10.75% after 48 h (Figure 3a). SW10 cells treated with VCR showed a cell death rate of 11.53 ± 2.01% after 24 h and 38.84 ± 2.71% after 48 h, respectively. The combination with NIM reduced cell death to a rate of 10.78 ± 0.94% after 24 h and 24.13 ± 2.26% after 48 h (Figure 3b). Statistical analysis data are shown in Appendix A for the RN33B cell line and Appendix A for the SW10 cell line.

### 2.3. NIM Pretreatment Increased Susceptibility of Tumor Cells to VCR Application

The malignant cell lines A549, SAS and LN229 showed similar results in the vehicle and NIM treatment alone experiments. A549 cells exhibited 2.98 ± 0.10% cell death rate after 24 h and 4.04 ± 1.36% after 48 h in the controls. The NIM-treated samples showed comparable values with a cell death rate of 1.72 ± 0.33% after 24 h and 8.82 ± 1.09% after 48 h (Figure 4a). The SAS cells showed a cell death rate of 2.57 ± 0.63% after 24 h and 4.43 ± 2.26% after 48 h in the controls. In the NIM-treated samples, we detected cell death values of 3.63 ± 0.19% after 24 h and 6.65 ± 0.55% after 48 h (Figure 4b). The LN229 cells showed a cell death rate of 2.26 ± 0.15% after 24 h and 3.50 ± 0.68% after 48 h in the controls. The NIM-treated samples showed comparable values with a cell death rate of 3.50 ± 0.14% after 24 h and 5.83 ± 0.12% after 48 h (Figure 4c). Treatment with VCR also showed a significant increase in cell death in the malignant cell lines. However, the combination with NIM intensified these effects. A549 cells treated with VCR showed a cell death rate of 2.81 ± 0.03% after 24 h and 34.67 ± 2.31% after 48 h, respectively. The combination with NIM increased the cell death rate to 8.51 ± 1.30% after 24 h and 45.49 ± 10.88% after 48 h (Figure 4a). SAS cells treated with VCR showed a cell death rate of 5.01 ± 0.28% after 24 h and 9.13 ± 0.85% after 48 h, respectively. The combination with NIM increased the cell death rate to 7.32 ± 0.06% after 24 h and 13.56 ± 0.91% after 48 h (Figure 4b). LN229 cells treated with VCR showed a cell death of 12.30 ± 1.37% after 24 h and 50.22 ± 0.86% after 48 h, respectively. The combination with NIM increased the cell death rate to 17.68 ± 1.18% after 24 h and 58.42 ± 0.62% after 48 h (Figure 4c). Statistical analysis data are shown in Appendix A for the A549 cell line, Appendix A for the SAS cell line, and Appendix A for the LN229 cell line.

### 2.4. NIM-Induced Effects Are Associated with Altered Protein Levels of AKT and CREB

The healthy neuronal or Schwann cells and the cancer cell lines were treated with 1 or 5 µM VCR with or without co-application of NIM. The transcription factors STAT (signal transducers and activators of transcription), AKT, ERK (extracellular signal-regulated kinases), CREB and LMO4 (LIM domain only 4) were analyzed by Western blot. The phosphorylated variants of each of STAT, AKT, ERK and CREB were also examined. GAPDH was used as a loading control. In the expression of STAT, the total amount was increased in RN33B and decreased in SW10 after treatment with VCR compared to the control. Co-treatment with NIM did not affect expression levels. The phosphorylated variant of STAT showed reduced protein levels under treatment, whereas NIM had no effect (Figure 5). The total amount of AKT was almost unchanged under treatment. Both cell lines showed increased phosphorylation of AKT at serine residue 473 with co-application of VCR and NIM compared to treatment with VCR alone (Figure 5). In both the RN33B and SW10 cells, co-application showed increased phosphorylation of CREB at serine residue 133. In each of the treated groups, the total amount of CREB was the same. (Figure 5). The total amount of ERK was reduced by VCR treatment, but the phosphorylated variant was homogeneously expressed in all groups. NIM treatment had no effect (Figure 5). For LMO4, a reduction in expression was observed under treatment compared to the control. Co-treatment with NIM did not result in any change in expression (Figure 5).

Quantification of the Western blots of pAKT and pCREB in RN33B and SW10. The protein signals were quantified using the ImageQuant TL software version 3.0 and normalized to the corresponding GAPDH signals. The treated samples were further normalized to the untreated control. For AKT, a reduced activation level was observed in RN33B treated with VIN (5.55 ± 0.76) compared to the combination of VIN and NIM (9.71 ± 0.87; *p* = 0.06) (Figure 6). Also, for the intensity of pCREB, RN33B showed a less strong signal with mono-treatment with VIN (2.30 ± 0.45) than in comparison with the combination of VIN and NIM (6.20 ± 0.80; *p* = 0.05) (Figure 6).

Similar trends were found for pAKT in SW10. Monotherapy with VIN resulted in a lower signal intensity (2.84 ± 0.48) compared to the combination of VIN and NIM (4.94 ± 0.75; *p* = 0.14) (Figure 6). For pCREB in SW10 cells, the signal intensity was reduced when treated with VIN (1.07 ± 0.01) compared to the treatment of VIN and NIM in combination (2.13 ± 0.02; *p* < 0.005) (Figure 6). The quantification of AKT, CREB, pSTAT and STAT is shown in Appendix A.

The same transcription factors were also measured in the tumor cell lines. For STAT, there was a discrete reduction in the total amount under VCR treatment in A549 compared to the control with no effect of co-treatment with NIM. The total amount of STAT was unchanged in SAS and LN229. Co-treatment with NIM did not lead to any change in protein levels in A549 and LN229 of the phosphorylated STAT. In SAS cells, co-treatment with NIM resulted in reduced protein levels of a phosphorylated variant of STAT (Figure 7). In contrast to the healthy cells, the cancer cell lines showed reduced phosphorylation of AKT with co-application of VCR and NIM compared to treatment with VCR alone. The total amount of AKT did not change in these cell lines (Figure 7). Similar to healthy cells, increased phosphorylation of CREB was observed with co-application of VCR and NIM. The total amount of CREB was reduced in A549 after VCR treatment, with no effect of NIM. The total amount of CREB was unchanged in all treated SAS and LN229 cells (Figure 7). For the total amount and the phosphorylated variant of ERK, there was no change in protein levels with VCR or co-application with NIM in all cell lines (Figure 7). LMO4 showed a slight reduction in expression with VCR treatment compared to the control. NIM had no further effect on expression levels (Figure 7). GAPDH protein levels were used as a loading control. The bands for pAKT and pCREB of the cell lines shown here were quantified (Figure 8).

In the SAS cell line, the quantification of pAKT in monotherapy showed a stronger signal intensity under VIN (3.66 ± 0.59) than in the combination of VIN with NIM (1.37 ± 0.18; *p* = 0.06). For pCREB, a reduction in band intensity was detectable in this cell line in monotherapy with VIN (10.63 ± 2.19) compared to the combination of VIN and NIM (19.06 ± 2.98; *p* = 0.15) (Figure 8).

In the A549 cell line, pAKT under monotherapy with VIN showed a reduction in band intensity (3.28 ± 0.15) compared to monotherapy of VIN and NIM (1.06 ± 0.06; *p* = 0.005). For pCREB, a low band intensity was evident in this cell line with VIN treatment (2.80 ± 0.60) compared to the combination of VIN and NIM (11.22 ± 0.43; *p* = 0.007) (Figure 8).

The glioblastoma cell line LN229 showed an increased signal (1.37 ± 0.07) regarding the band intensity of pAKT under monotherapy with VIN compared to the combination of VIN and NIM (0.57 ± 0.14; *p*= 0.03). For pCREB, a low band intensity (4.28 ± 0.29) was also detected in this cell line under therapy with VIN compared to the combination of VIN and NIM (5.57 ± 0.43; *p* = 0.13) (Figure 8). The quantification of AKT, CREB, pSTAT and STAT is shown in Appendix A.

To visualize the results of each cell line with respect to the combination of VCR and NIM in terms of cell death as well as the protein levels of pAKT and pCREB, we designed Figure 9.

## 3. Discussion

One of the main problems with numerous chemotherapeutic agents is the dose-limiting effect caused by damage to healthy cells and the development of resistance in tumor cells [17,20,21,22,29,30]. For example, neuropathy and even blindness are often a side effect of VCR therapy [13,14,31]. Ideally, the aim is to protect healthy tissue from the chemotherapeutic agent and improve the response of tumor cells. In this study, we have shown that NIM protects neuronal and Schwann cells from VCR and reduces cell death in these cell lines. At the same time, it increased the sensitivity of the tumor cell lines used to VCR.

Many substances have already been investigated in preclinical studies to prevent VCR-induced neuropathy. Vitamin B6 (pyridostigmine) or glutamate, for example, have shown promising results [18]. Glutamate also showed neuroprotective properties in VCR-induced neuropathy in other studies [32]. However, this has not yet been used in routine clinical practice. In a study by Helleputte et al., inhibition of histone deacetylase 6 (HDAC6) was able to reduce VCR-induced neuropathy and also reduce tumor growth in the mouse model for acute lymphoblastic leukemia [33]. A limiting factor, unfortunately, is often the translation of preclinical data into clinical routine.

One substance that is already regularly used in patient care due to its neuroprotective properties is NIM. The effect was mainly analyzed in the surgical treatment of vestibular schwannomas with regard to hearing preservation [26]. Preclinical investigations by our research group were able to show that NIM has these neuroprotective properties under mechanical, osmotic and heat stress in a model with neuronal and Schwann cells [25,34].

In addition to its neuroprotective effects, other preliminary studies have shown that NIM has a sensitizing effect on tumor cells undergoing chemotherapy. NIM showed promising results in preclinical trials in malignant glioma cells. Durmaz et al. demonstrated that the therapy effects of VCR, carmustine and procarbazine on glioblastoma cells were increased with NIM but not with verapamil [35]. Kiwit et al. studied the effect of malignant glioma cells under treatment with (1,4-amino-2-methyl-5-pyrimidinyl)-methyl-3-(2-chloroethyl)-3-nitrosoure a (ACNU), VCR and cisplatin with verapamil and NIM. Resistance to chemotherapy could be overcome by the addition of these calcium channel inhibitors [36]. Studies by Kunert-Radek et al. showed concentration-dependent antiproliferative effects of NIM and verapamil on glioblastoma cells. However, concentrations in the millimole range were used here, i.e., significantly higher than in our study [37]. A stronger response to therapy was also seen in neuroblastoma cells when VCR was combined with amlodipine [38]. NIM also showed similar effects in other tumor entities. In a model with melanoma cells, a reduction in tumor cell aggregation, colony formation and metastasis were observed [39].

Based on the data from the Western blot, we were able to determine an increased protein levels of pAKT in the benign cells. In contrast, we found opposing effects in the malignant cells with a reduction in the phosphorylation of AKT. As in previous studies, the neuroprotective effects of the healthy cells were associated with an increase in pCREB. However, the increased protein level was also evident in the tumor cells, where treatment with VCR and NIM led to an increase in cell death.

In our own preliminary work, we showed that the neuroprotective effect of NIM in neuronal cells was associated with the upregulation of AKT and CREB [25]. Hu et al. also found that activation of AKT and CREB was associated with NIM-dependent neuroprotection [40]. Furthermore, activation of AKT was associated with protection against VCR-associated nerve damage in a rat model [41]. In a study by Kuskabe et al., a neuroprotective effect of NIM was observed in PC12 cells, which was associated with activation of CREB as well as ERK and upregulation of Ca-binding proteins such as calmodulin, calbindin and calretinin [42]. Along with CREB, AKT is considered one of the most important signaling pathways for neuronal survival in the context of ischemia [43]. Molina-Salinas’ study showed in a rat model that neuronal survival in the presence of toxins is mainly mediated by the activation of AKT [44].

However, AKT is also overactivated in many tumors and is important for tumor cell survival [45,46,47]. It plays a critical role in tumor resistance and proliferation and role in DNA damage response and repair [48]. For example, overexpression of AKT is associated with resistance to VCR in various tumors, such as colorectal carcinoma or lung cancer [49,50,51]. In addition, activation of AKT by erythropoietin was linked to a resistance against VCR in neuroblastoma cells [52].

On the other side, inhibition of AKT was able to overcome chemoresistance in ovarian cancer, breast cancer and colorectal carcinoma [53,54]. The study by Gao et al. also showed that inhibition of both AKT and ERK in oral carcinoma cells led to a reduction in migration and invasion [47]. For this reason, some authors see specific inhibitors of AKT as a possible therapy for tumor diseases [46]. In the future, we also plan to conduct experiments with AKT inhibitors to gain further insight into the molecular mechanisms involved.

In parallel with the activation of AKT, the neuroprotective effects in this study were also associated with increased phosphorylation of CREB. The AKT, ERK and CREB signaling pathways have been identified as promising targets for neuroprotection, for example, in stroke [55,56].

In a rat model, improved nerve regeneration by NIM after trauma was also associated with activation of CREB [57]. The protective effects of NIM in a mouse model of subarachnoid hemorrhage were associated with increased phosphorylation of CREB and AKT in the frontal cortex, while ERK remained unchanged [58]. In addition to its role in neuroprotection, CREB also plays a role in neuronal plasticity [59].

On the other hand, CREB is also considered a proto-oncogene in many tumors, promoting tumor initiation, progression and metastasis. CREB is often overexpressed and overactivated in tumors such as non-small-cell lung cancer (NSCLC) or acute myeloid leukemia (AML) [60,61,62]. CREB levels correlate with tumor stage in gastric cancer, for example, and are thought to be important in the development of leukemia [63,64]. In addition, the CREB level in the blood can be indicative of the risk of recurrence in AML [62]. Therefore, as potential therapeutic agents, CREB inhibitors are also of interest [60]. To further investigate the observed effects, further experiments with such inhibitors are planned.

The transcription factor LMO4 plays an important role in cell cycle regulation and apoptosis [65]. Overexpression of LMO4 appears to be associated with more aggressive behavior and poorer survival in many cancers, like gastric cancer cells, breast cancer and NSCLC [66,67,68]. Wang et al. also showed that LMO4 leads to increased migration and invasion via activation of AKT [67,68]. Another study from our laboratory also demonstrated that NIM can protect auditory hair cells from cisplatin-induced cell death. These effects were associated with an upregulation of the transcription factor LMO4 [28]. In our experiments, we saw a reduction in LMO4 by VCR treatment but no effect by the simultaneous administration of nimodipine.

Other studies demonstrated that ERK and CREB are associated with sensitivity to VCR in leukemic cells. Here, dexamethasone treatment increases the phosphorylation of CREB and reduces the phosphorylation of ERK [69]. However, ERK does not appear to play a role in the effects observed here.

Another important player in tumorigenesis is STAT [70,71]. Overexpression of STAT is associated with more aggressive behavior in many tumors [72,73,74]. Inhibitors of the JAK (janus kinase)/STAT signaling pathway show promising results for some cancer types, but this pathway is also important for an immune response [71,73,75]. STAT3 has been linked to inflammatory changes in tumor microenvironment and to cancer cell survival [71,74]. In our study, we saw a partial change in the expression of STAT as a result of the treatment but without any effect of NIM.

### Limitations and Outlook

So far, we have been able to show that nimodipine enhances the efficacy of VCR in various tumor cells while protecting neuronal cells. However, the data to date are rather descriptive. Despite the correlation with the differential protein levels of pAKT and pCREB, the data do not yet allow conclusions about causal relationships or mechanisms. Therefore, further experiments are necessary, e.g., with inhibitors of AKT and CREB. Furthermore, the cell lines used are not suitable to map the intra- and intertumoral heterogeneity of, e.g., glioblastomas. Therefore, experiments with primary cultures and patient-derived organoids are planned as the next steps. In addition, the typical first-line therapies, such as Temodal for glioblastomas, will also be investigated in combination with NIM.

## 4. Materials and Methods

### 4.1. Cell Lines

The cancer cell lines A549 (human non-small cell lung cancer, NSCLC) and SAS (squamous tongue cancer) were kindly provided by Barbara Seliger (Institute of Medical Immunology, Martin Luther University Halle-Wittenberg, Halle (Saale), Germany). The cancer cell line LN229 (human glioma, #305043) was acquired from Cell Lines Service (Eppelheim, Germany). The neuronal mammal cell lines RN33B (CRL-2825, neurons, rat) and SW10 (CRL-2766, Schwann cells, mouse) were acquired from the ATCC (American Type Culture Collection, Manassas, VA, USA).

All cell media were purchased from Thermo Fisher Scientific, Waltham, MA, USA, and supplemented with 10% FBS (Fetal Bovine Serum, Thermo Fisher Scientific, Waltham, MA, USA) and 1% penicillin–streptomycin (10,000 units/mL penicillin, 10,000 μg/mL streptomycin, Thermo Fisher Scientific, Waltham, MA, USA). LN229 cell line was cultured in Gibco RPMI 1640, A549, SAS, SW10 cell lines in Gibco 1× DMEM and RN33B cell line in Gibco DMEM F12 1:1 Medium. Cells were cultured in 75 cm² cell culture flasks (Sarstedt, Nümbrecht, Germany) in an incubator at 37 °C in a humid atmosphere with 5% CO_2_ concentration.

### 4.2. NIM and VCR Treatment

A total of 5 × 10^4^ cells of each cell line were seeded in 24-well plates (Techno Plastic Products, TPP, Trasadingen, Switzerland). Approximately 24 h prior to VCR application, the cells were treated with 20 µM NIM diluted in absolute ethanol (EtOH). Equal amounts of EtOH were added to non-treated controls (0.1% final concentration, vehicle). The NIM solutions and the treated cells were protected from light. Afterwards, VCR was added to each cell line to receive concentrations of 1 µM and 5 µM. An overview of the treatment scheme and the experimental set-up is given in Figure 10.

### 4.3. Cytotoxicity Measurement

After 24 h and 48 h, cytotoxicity was measured by the lactate dehydrogenase activity as a marker for cell death using Cytotoxicity Detection Kit (#11644793001, Roche, Basel, Switzerland) according to the manufacturer’s instructions. In brief, 100 µL cell culture supernatant in triplicates per sample and 100 µL reaction mix were incubated in the dark for 30 min. Absorbance was measured at 492 nm with Tecan Reader F2000 Pro (Tecan, Männedorf, Switzerland) at four definite points of the wells. The absorbance of cells lysed with 2% Triton X-100 (Carl Roth, Karlsruhe, Germany) served as a positive control (100% cell death), while the medium signal without cells served as the background signal. The calculation of the cell death rate was performed as described before [25]. The diagrams show the means and standard deviations (SD) of triplicates from one representative assay out of three biological replicates.

### 4.4. Fluorescence Microscopy

Cells were seeded at a density of 5 × 10^4^ cells per well in 24-well plates (Techni Plastic Products, TPP, Transdingen, Switzerland). The cells were then treated with 1 µM VCR, either with or without pretreatment, using 20 µM NIM diluted in absolute ethanol. After, 24 h microscope imaging was performed. For cell staining, CellRox™ (#C10444, Thermo Fisher Scientific, Waltham, MA, USA) and NucBlue™ Live Cell Stain ReadyProbes reagent (#R37605, Hoechst 33342, Thermo Fisher Scientific, Waltham, MA, USA) were used according to the manufacturer’s instructions. After staining, cells were washed with Dulbecco’s phosphate-buffered saline (PBS, Gibco, Thermo Fisher Scientific, Waltham, MA, USA) and overlaid with FluoroBrite™ DMEM (Gibco, Thermo Fisher Scientific, Waltham, MA, USA) for imaging. Imaging was performed with a Keyence BZ-800E microscope (Keyence, Neu-Isenburg, Germany). The quantification of cells stained with NucBlue™ and CellROX™ was performed using the IdentifyPrimaryObjects function in the CellProfiler software (Version 4.2.4, Broad Institute, Cambridge, MA, USA).

### 4.5. Western Blot

Cells were seeded on 96 mm × 21 mm tissue culture dishes (Techno Plastic Products, TPP, Trasadingen, Switzerland) and pretreated with either 20 µM NIM diluted in absolute ethanol or the same amount of absolute ethanol without NIM in the control cultures. After 24 h, either 1 µM or 5 µM VCR was added to the cells. After a further 24 h, the cells were washed with ice-cold PBS (Thermo Fisher Scientific, Waltham, MA USA) and harvested using cold PBS diluted Pierce™ Protease Inhibitor Mini Tablet, EDTA-free (#A32961, Thermo Fisher Scientific, Waltham, MA, USA) and centrifuged (300 g/5 min/4 °C). For cleavage of nucleic acid bonds, Benzonase-Nuclease (Merck, Sigma Aldrich, St. Louis, MO, USA) was used. After extraction with Invitrogen 1× LDS sample buffer (#B0007, Thermo Fisher Scientific, Waltham, MA, USA), the proteins were heated at 70 °C for 10 min. Protein concentration was measured using the Pierce™ BCA Protein Assay Kits (#23227, Thermo Fisher Scientific, Waltham, MA USA). Proteins at the correct concentration were diluted in 5% β-mercaptoethanol (Carl Roth, Karlsruhe, Germany) and 1× LDS sample buffer and again heated at 70 °C for 10 min.

To separate the proteins, sodium dodecyl sulphate-polyacrylamide gel electrophoresis was performed using NuPAGE™ 4–12%, Bis-Tris, 1.5 mm, Mini-Protein-Gels (#NP0335BOX, Thermo Fisher Scientific, Waltham, MA USA) and NuPAGE™ MES SDS Running Buffer (Thermo Fisher Scientific, Waltham, MA USA). Proteins were then blotted onto nitrocellulose membranes using iBlot™ 2 Transfer Stacks (#IB23001, Thermo Fisher Scientific, Waltham, MA USA) and the iBlot™ 2 Dry Blotting System (Thermo Fisher Scientific, Waltham, MA USA). The membranes were then stained with Ponceau S (0.1% Ponceau S, 3% trichloroacetic acid and 3% sulfosalicylic acid).

To block the proteins on the membrane, 5% skim milk powder (Carl Roth, Karlsruhe, Germany) diluted in Tris-buffered saline with 0.1% Tween (TBS-T, Sigma-Aldrich, St. Louis, MO, USA) was applied for 1 h.

The membranes were incubated overnight at 4 °C with added primary antibodies (see Table 1). On the next day, three washes with TBS and two further washes with TBS-T were performed. The secondary antibody was then added for 1 h at room temperature.

The blots were developed using the SuperSignal West Femto Chemiluminescent Substrate (#34580, Thermo Fisher Scientific, Waltham, MA, USA), and the signals were captured using a CCD camera (ImageQuant LAS4000, GE Healthcare, Freiburg, Germany). GAPDH (glyceraldehyde-3-phosphate dehydrogenase) was utilized as a loading control for the proteins. Protein bands were quantified using ImageQuant TL software version 3.0 (GE Healthcare, Freiburg, Germany), and the results were subsequently normalized to GAPDH.

### 4.6. Statistical Analysis

All data analyses were performed using Microsoft Excel software (version Microsoft Office Professional Plus 2016, Microsoft Corporation, Redmond, WA, USA) and GraphPad Prism 4.9.1 (GraphPad Software Inc., SanDiego, CA, USA). Tukey’s multiple comparison test was used for the analysis of LDH assays. For quantification of Western blots, an unpaired two-tailed Student’s *t*-test was used. Figures are mean or standard deviation (SD). For each experiment, at least three biological replicates were performed.

## 5. Conclusions

To conclude, NIM protects neuronal cells from VCR-associated cell death. It was striking that there was an increased expression of pAKT in healthy cells, which was associated with neuroprotection. On the other hand, a reduced expression of pAKT was evident in tumor cells, which was associated with a stronger response to chemotherapy.

By improving tumor cell response to chemotherapy while protecting normal tissue, survival can be improved and adverse effects, like neuropathy, reduced (Figure 11). Further studies like animal models are needed to validate our data.

## Figures and Tables

**Figure 1 ijms-25-10389-f001:**
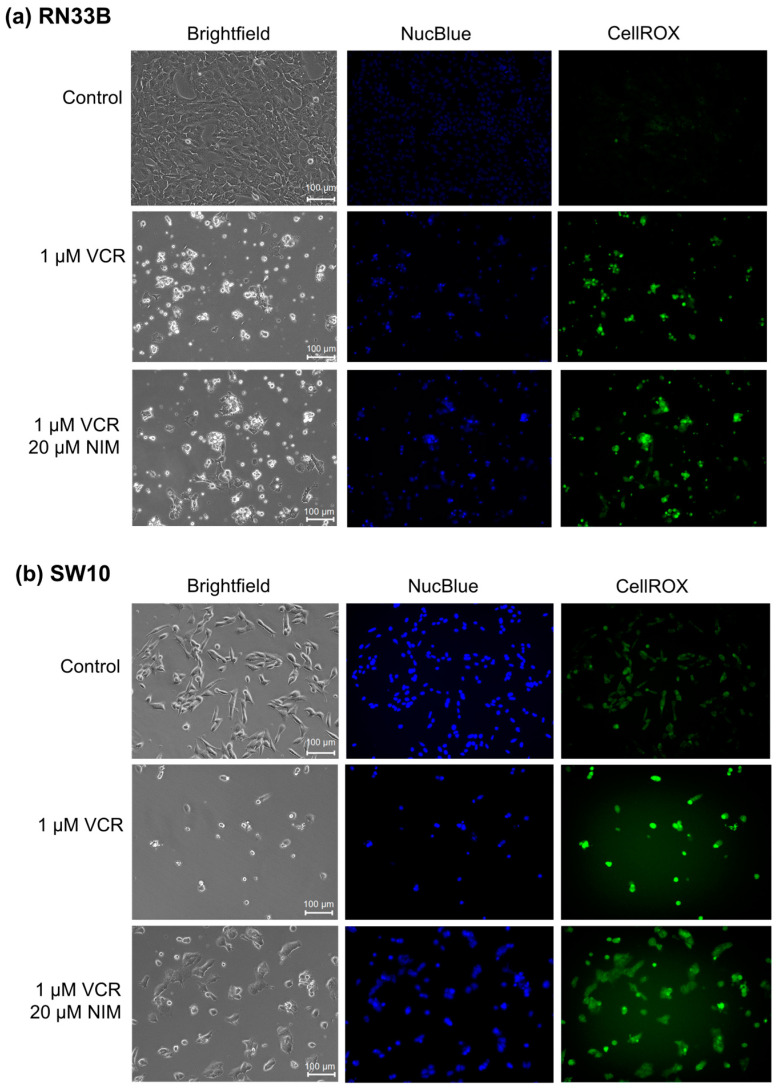
Microscopic images of RN33B (**a**) and SW10 (**b**) cells with 100× magnification. Top row shows control. Middle row shows cells after 24 h treatment with VCR. Bottom row shows cells after 24 h pretreatment of cells with NIM and subsequent addition of VCR. Brightfield (first column) and fluorescence microscopic images with NucBlue^™^ staining (blue, middle column) and CellROX^™^ (green, right column). Scale bar = 100 µm.

**Figure 2 ijms-25-10389-f002:**
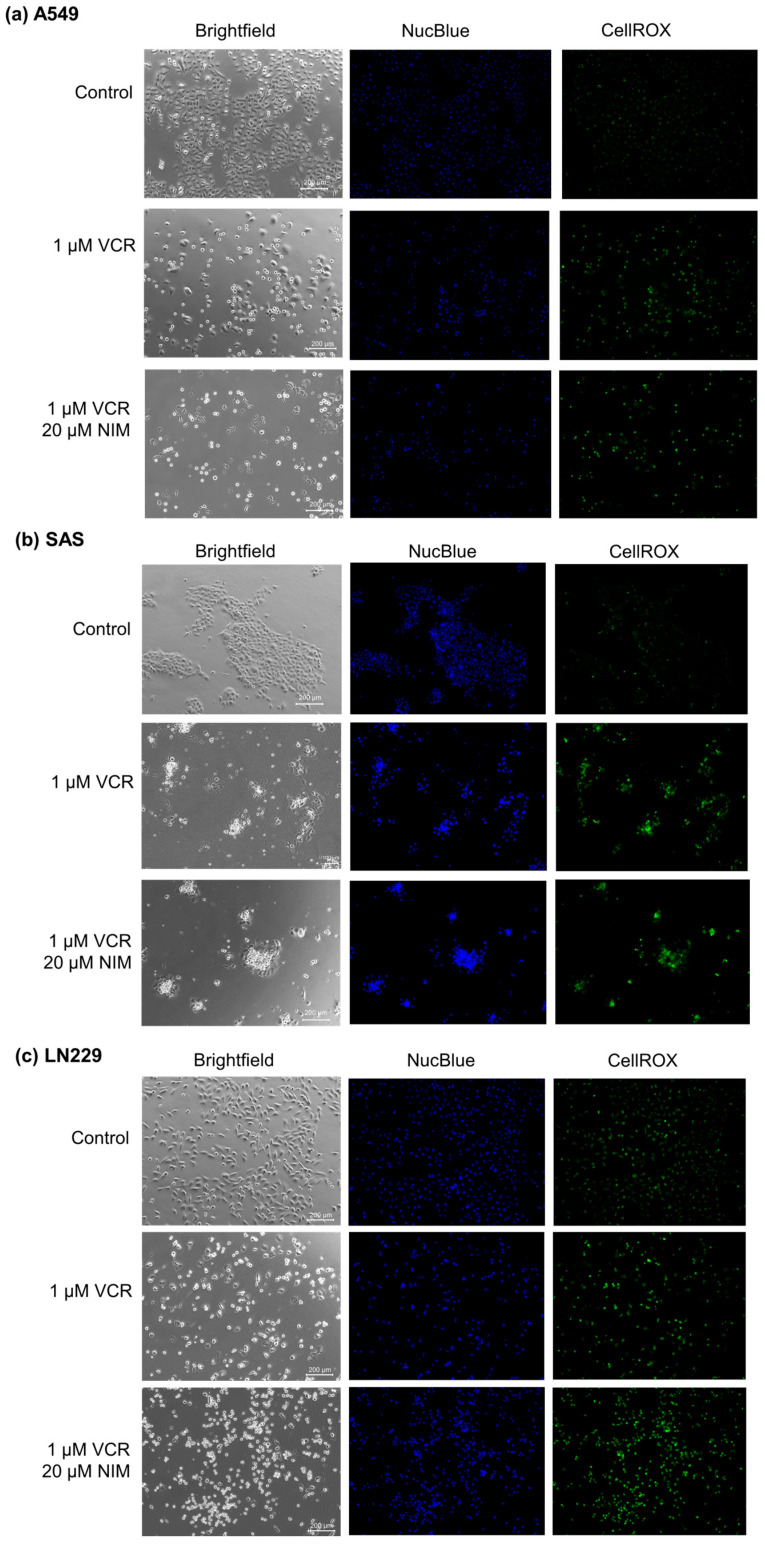
Microscopic image of A549 (**a**), SAS (**b**) and LN229 (**c**) cells with 100× magnification. Top row shows control. Middle row shows cells after 24 h treatment with VCR. Bottom row shows cells after 24 h pretreatment of cells with NIM and subsequent addition of VCR. Brightfield (left column) and fluorescence microscopic images with NucBlue™ (Hoechst 33342) staining (blue, middle column) and CellROX^™^ (green, right column). Scale bar = 200 µm.

**Figure 3 ijms-25-10389-f003:**
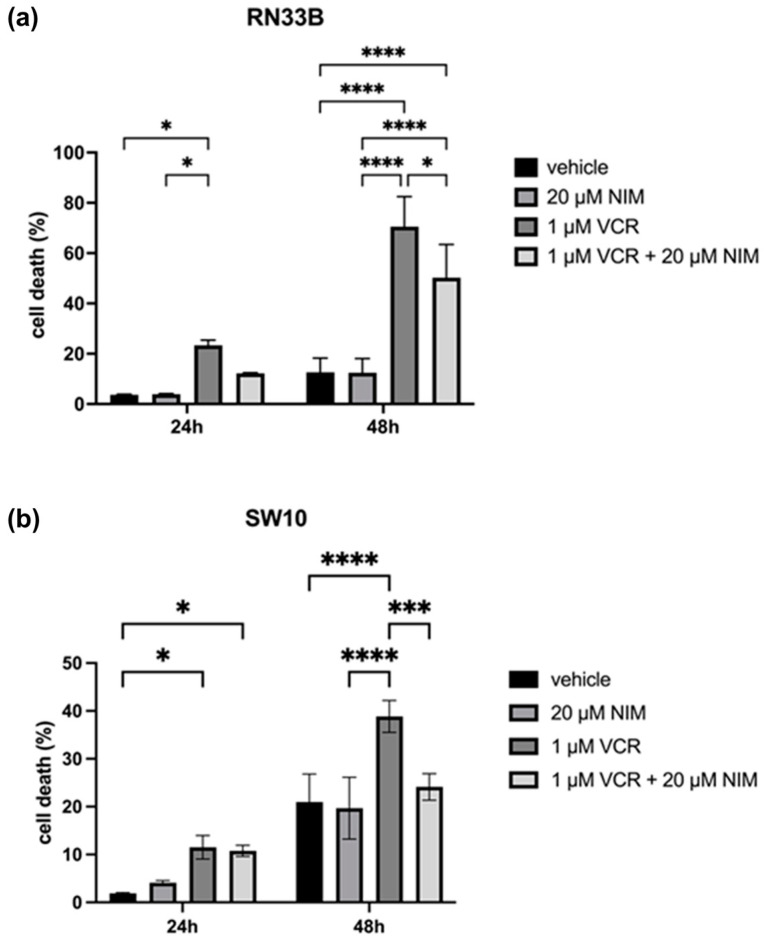
Cell viability of RN33B (**a**) and SW10 (**b**) cells measured after 24 h and 48 h by LDH assay. The viability was calculated using a sample treated with Triton X100 (cell death = 100%). The vehicle (ethanol) and the NIM-pretreated samples are compared. The diagrams show the mean values and SDs of three independent biological replicates. Tukey’s multiple comparisons test was performed for statistical analysis. * *p* < 0.05; *** *p* < 0.001; **** *p* < 0.0001.

**Figure 4 ijms-25-10389-f004:**
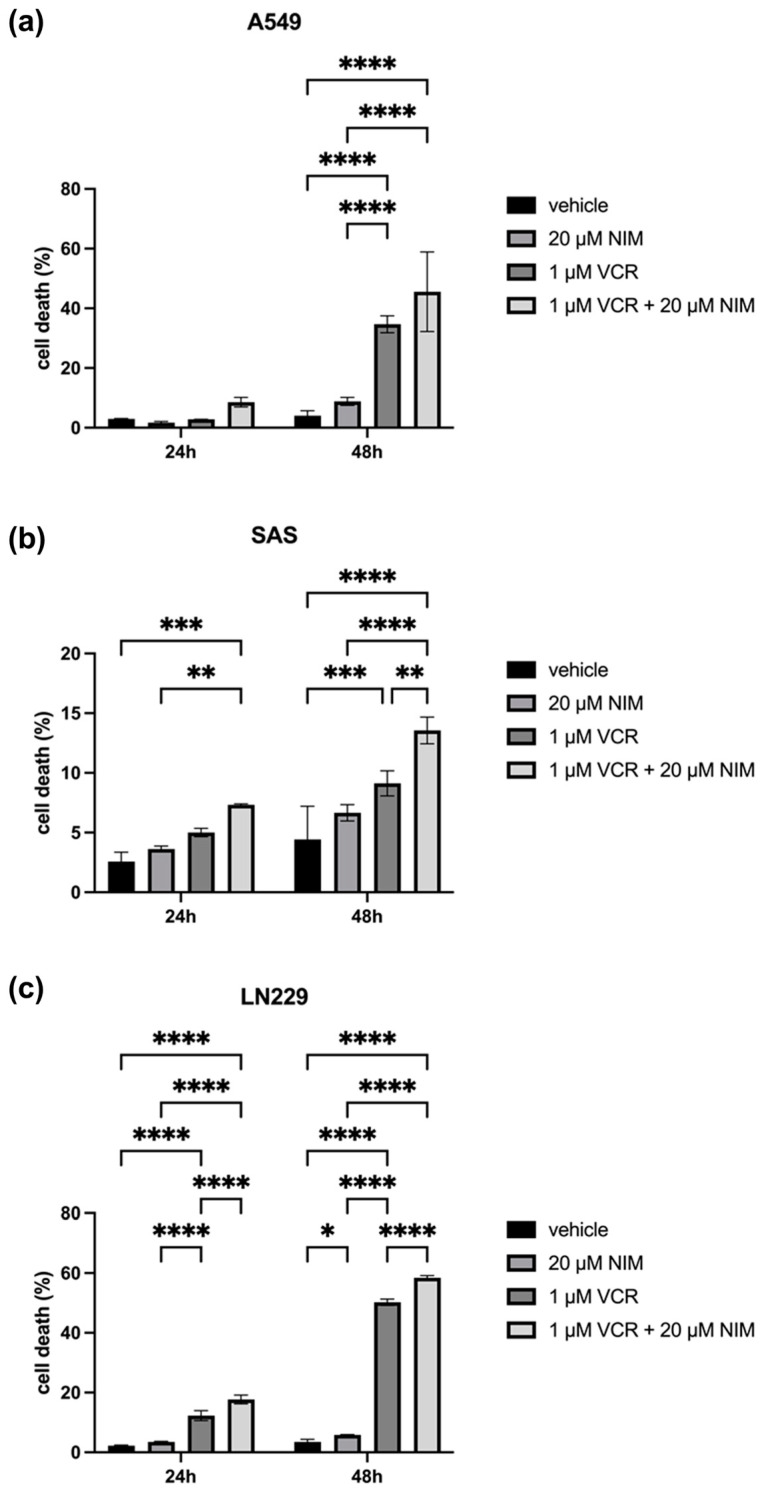
Cell viability of A549 (**a**), SAS (**b**) and LN229 (**c**) cells measured after 24 h and 48 h by LDH assay. The viability was calculated using a sample treated with Triton X (cell death = 100%). The data for the samples without stress are shown on the left and the samples with VCR treatment on the right. The vehicle (ethanol) and the NIM-pretreated samples are compared. The diagrams show the mean values and SDs of three independent biological replicates. Tukey’s multiple comparisons test was performed for statistical analysis. * *p* < 0.05; ** *p* < 0.01; *** *p* < 0.001; **** *p* < 0.0001.

**Figure 5 ijms-25-10389-f005:**
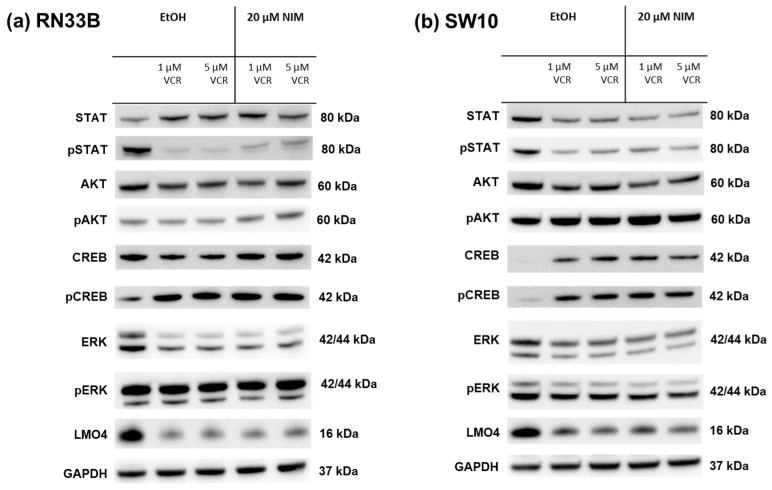
Detection of transcription factors in neuronal cells RN33B (**a**) and SW10 (**b**) treated with VCR alone as well as with co-application of NIM. After the transfer of the proteins separated by SDS-PAGE onto nitrocellulose membranes, phosphorylation and total protein levels of the cell signaling components were determined by specific antibodies. The GAPDH protein level was used as a loading control. The Western blot shown is representative of the results from three independent biological replicates.

**Figure 6 ijms-25-10389-f006:**
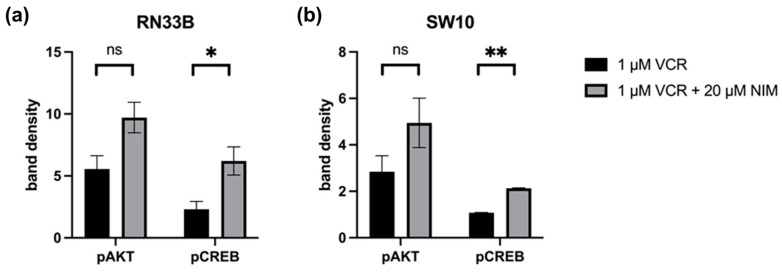
Quantification of pAKT and pCREB. Quantification of the bands in RN33B (**a**) and SW10 (**b**) was normalized to the GAPDH control. The mean values including standard deviation are shown. Two independent biological replicates were analyzed. Student’s *t*-test was performed for statistical analysis. * *p* < 0.05; ** *p* < 0.01; ns: not significant.

**Figure 7 ijms-25-10389-f007:**
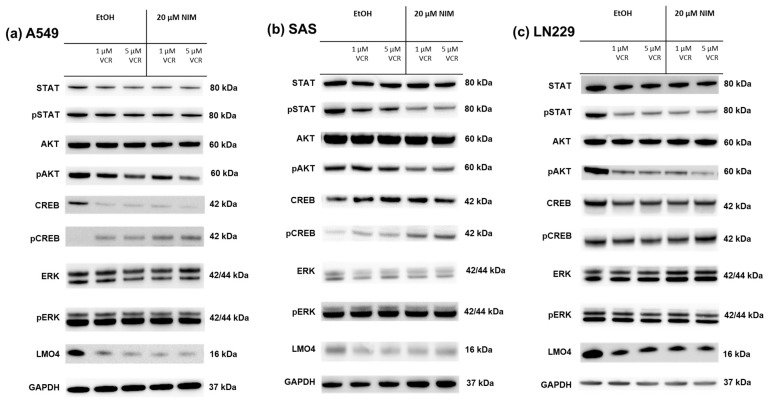
Detection of transcription factors in cancer cell lines A549 (**a**), SAS (**b**) and LN229 (**c**) treated with VCR alone as well as with co-application of NIM. After the transfer of the proteins separated by SDS-PAGE onto nitrocellulose membranes, phosphorylation and total protein levels of the cell signaling components were determined by specific antibodies. The GAPDH protein level was used as a loading control. The Western blot shown is representative of the results from three independent biological replicates.

**Figure 8 ijms-25-10389-f008:**
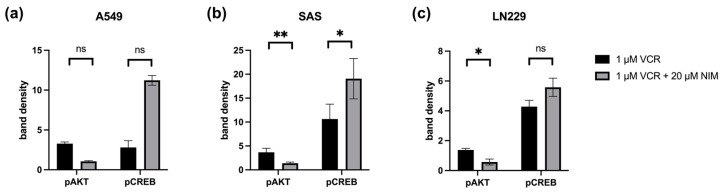
Quantification of pAKT and pCREB in A549 (**a**), SAS (**b**) and LN229 (**c**). Quantification of the bands normalized to the GAPDH control. The combination treatment was also normalized to the monotherapy. The mean values including deviation are shown. Two independent biological replicates were analyzed. Student’s *t*-test was performed for statistical analysis. * *p* < 0.05; ** *p* < 0.01; ns: not significant.

**Figure 9 ijms-25-10389-f009:**
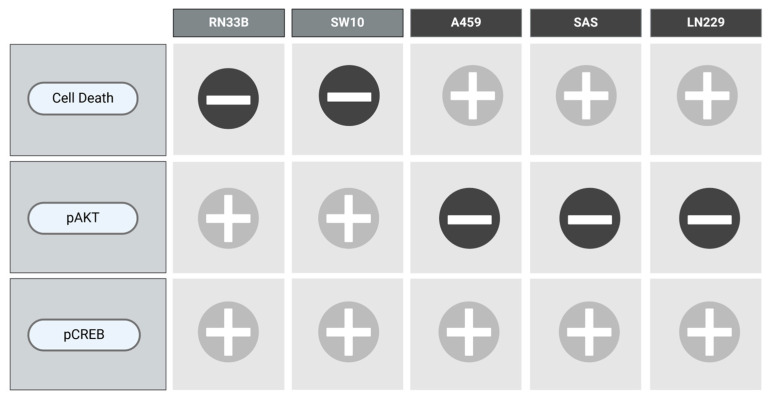
Overview of the main results.

**Figure 10 ijms-25-10389-f010:**
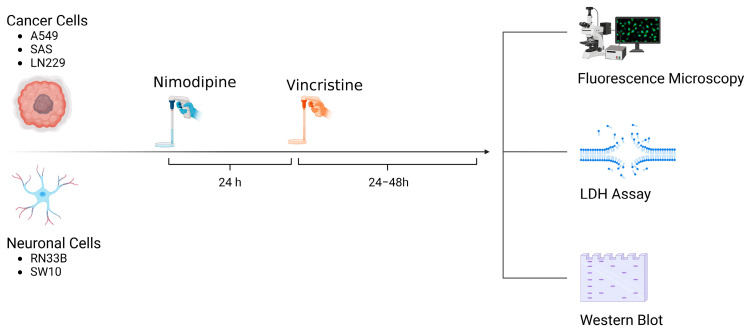
Overview of treatment scheme and experimental set-up (Created with BioRender^®^, Agreement number: WR273JJ9DL).

**Figure 11 ijms-25-10389-f011:**
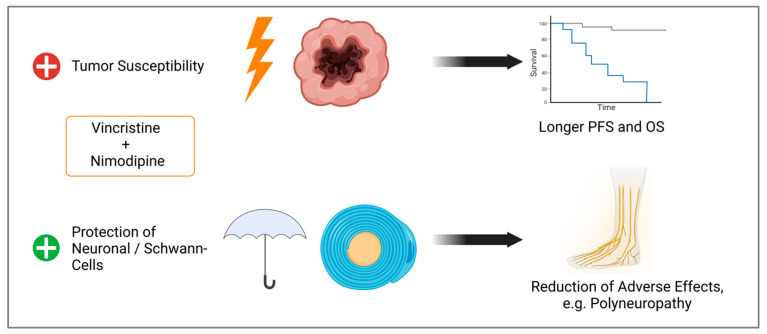
Observed effects of the combination of VCR and NIM. Improved treatment response of tumor cells, which could lead to longer progression free survival (PFS) and overall survival (OS). Simultaneously, we demonstrated protection against chemotherapy-associated cell death of Schwann and neuronal cells, which could lead to a reduction in adverse effects (Created with BioRender^®^, Agreement number: N273JM90A).

**Table 1 ijms-25-10389-t001:** List of used antibodies.

Antibody	Species	Dilution	Dilution Buffer	Manufacture
AKT (40D4) #2920	Mouse IgG1	1:2000	5% MP in TBS-T	Cell Signaling Technology (Danvers, MA, USA)
Phospho-Akt (Ser473) (D9E) #4060	Rabbit IgG	1:1000	5% BSA in TBS-T	Cell Signaling Technology (Danvers, MA, USA)
CREB (48H2) #9197	Rabbit IgG	1:1000	5% BSA in TBS-T	Cell Signaling Technology (Danvers, MA, USA)
Phospho-CREB (Ser133) (87G3) #9198	Rabbit IgG	1:1000	5% MP in TBS-T	Cell Signaling Technology (Danvers, MA, USA)
ERK 1/2 (T202/ Y204) #9102	Rabbit IgG	1:1000	5% BSA in TBS-	Cell Signaling Technology (Danvers, MA, USA)
Phospho-ERK 1/2 (Thr202/Tyr204) #9101	Rabbit IgG	1:1000	5% BSA in TBS-T	Cell Signaling Technology (Danvers, MA, USA)
LMO4 (D6V4Z) #81428	Rabbit IgG	1:1000	5% BSA in TBS-T	Cell Signaling Technology (Danvers, MA, USA)
GAPDH (14C10) #2118	Rabbit IgG	1:1000	5% BSA in TBS-T	Abcam (Cambridge, UK)
STAT3 (124H6) #9139	Mouse IgG2a	1:1000	5% MP in TBS-T	Cell Signaling Technology (Danvers, MA, USA)
Phospho-STAT3 (Tyr705) (3E2) #9138	Mouse IgG1	1:1000	5% MP in TBS-T	Cell Signaling Technology Inc. (Danvers, MA, USA)
Anti-Rabbit IgG, HRP-linked Antibody #7074	Goat	1:1000	2% MP in TBS-T	Cell Signaling Technology Inc. (Danvers, MA, USA)
Anti-Mouse IgG, HRP-linked Antibody #7076	Horse	1:1000	2% MP in TBS-T	Cell Signaling Technology Inc. (Danvers, MA, USA)

Abbreviations: AKT, protein kinase B; BSA, bovine serum albumin; CREB, cAMP response element-binding protein; GAPDH, glyceraldehyde-3-phosphatedehydrogenase; ERK, extracellular signal-regulated kinases; HRP, horseradish peroxidase; IgG, immunoglobulin G; LMO4, LIM domain only 4; MP, milk powder; Stat, signal transducer and activator of transcription.

## Data Availability

The dataset is available from the corresponding author upon reasonable request.

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
