# Peer review of "Nimodipine Used with Vincristine: Protects Schwann Cells and Neuronal Cells from Vincristine-Induced Cell Death but Increases Tumor Cell Susceptibility"

_ijms, 2024, doi:10.3390/ijms251910389_

Round 1
Reviewer 1 Report
Comments and Suggestions for Authors
The authors investigate whether NIM has neuroprotective properties in neuronal cells during chemotherapy with VCR. Unfortunately it is not clear for this reviewer why did the authors decided to focus on NIM +VCR... More details need to be added in the Introduction.
After reading also the Discussion, it feels like a big part of the Discussion should be moved to Introduction, to justify this study...
Why did the authors focus on those 3 cancer cell lines? Any particular reason? Can the results be extended to other cancer cell lines?
Author Response
The authors investigate whether NIM has neuroprotective properties in neuronal cells during chemotherapy with VCR. Unfortunately it is not clear for this reviewer why did the authors decided to focus on NIM +VCR... More details need to be added in the Introduction.
- Answer 1: We would like to thank the reviewer for this comment. The rationale behind the combination of a VCR with a neuroprotective agent is elucidated in the introduction line from line 32. “However, the activity of VCR is not specific or limited to tumor cells. As a result, side effects such as polyneuropathy or cranial nerve deficits and even optic atrophy may occur due to damage to cells of the nervous system [12–16]. VCR-induced peripheral neuropathy affects up to 80% of patients and is the main dose-limiting toxicity [16, 17]. A substance used for neuroprotection in combination with VCR has not yet been established. To date, there are only a few preclinical studies that attempt to reduce the neurotoxicity of VCR [18, 19].” To explain the use of the combination of nimodipine and vincristine in more detail, we have slightly modified the introduction (page 2, line 43): "One compound that has shown neuroprotective properties in other settings is nimodipine (NIM)."
After reading also the Discussion, it feels like a big part of the Discussion should be moved to Introduction, to justify this study...
- Answer 2: We hope that by adapting the introduction, the meaning of the study will now be clearer. The beginning of the discussion is, in our point of view, important for understanding and should therefore be left as it is.
Why did the authors focus on those 3 cancer cell lines? Any particular reason? Can the results be extended to other cancer cell lines?
- Answer 3: In previous work of our research group, we were able to demonstrate neuroprotective properties of nimodipine under other questions or constellations (e.g. PMID: 35328753, PMID: 35628594, PMID: 25447789, PMID: 32283212, PMID: 25584831, PMID: 25258303). In order to investigate these effects also under chemotherapeutic stress by vincristine, this experimental setup was chosen and neuronal cells (investigation of neurotoxicity) as well as various tumor cells were treated with vincristine. In order to study these effects also under chemotherapeutic stress by vincristine, this experimental setup was chosen and neuronal cells (investigation of neurotoxicity), as well as different tumor cells treated with vincristine. These three lines were chosen because they are typical of the diseases for which vincristine can be used. Whether the results can generally be transferred to other tumor cell lines cannot be concluded based on the data of this study. However, we were able to demonstrate, for example, that NIM protects auditory cells also under cisplatin stress (PMID: 35628594).
Reviewer 2 Report
Comments and Suggestions for Authors
In this manuscript, the authors deal with a topic, from my point of view, of great social and health value. Co-treatment of cancer patients with a chemo-herapeutic (vincristine) and a calcium antagonist (Ni-modipine) protects against peripheral neuropathic damage typical of cancer patients after drug treatment. The authors medinate in vitro studies have shown that this combination of drugs maintains the chemotherapeutic efficiency towards tumour cells but protects nerve cells. All the methods used are suitable for this type of study, made even clearer with the help of figures showing treatment schemes. The manuscript in all its sections is well related and comprehensive. Only one objection is to insert the catalogue number to the kits used.
Author Response
In this manuscript, the authors deal with a topic, from my point of view, of great social and health value. Co-treatment of cancer patients with a chemo-herapeutic (vincristine) and a calcium antagonist (Ni-modipine) protects against peripheral neuropathic damage typical of cancer patients after drug treatment. The authors medinate in vitro studies have shown that this combination of drugs maintains the chemotherapeutic efficiency towards tumour cells but protects nerve cells. All the methods used are suitable for this type of study, made even clearer with the help of figures showing treatment schemes. The manuscript in all its sections is well related and comprehensive. Only one objection is to insert the catalogue number to the kits used.
Answer: We thank the reviewer for this suggestion. The catalogue numbers have been added.
Reviewer 3 Report
Comments and Suggestions for Authors
The authors of the manuscript submitted for review investigated the effects of the combination of vincristine and nimodipine on three cancer cell lines (A549, SAS and LN229) and neuronal cells (RN33B, SW10). The authors used fluorescence microscopy, LDH assays and Western blot analysis. From the results of the study, the authors concluded that Nimodipine was able to enhance the cell death effects of vincristine in cancer cells and at the same time protect neuronal cells. The conducted studies may improve the tumor response to chemotherapy and reduce neuropathy.
In my opinion the manuscript is valuable and interesting, well-structured and the research is well-conducted. After minor corrections it can be accepted for publication.
- The authors could include one of the new reviews on vincristine in the cited literature (e.g. Shukla R, Singh A, Singh KK. Vincristine-based nanoformulations: a preclinical and clinical studies overview. Drug Deliv Transl Res. 2024 Jan;14(1):1-16. doi: 10.1007/s13346-023-01389-6.),
- figure titles should be shorter and the descriptions contained in them should be moved to the text,
- in lines 39, 49 and 78 there are unnecessary dashes in words,
- a summary of the results of cell studies in the form of tables would allow for better visibility of the results obtained.
Author Response
- The authors could include one of the new reviews on vincristine in the cited literature (e.g. Shukla R, Singh A, Singh KK. Vincristine-based nanoformulations: a preclinical and clinical studies overview. Drug Deliv Transl Res. 2024 Jan;14(1):1-16. doi: 10.1007/s13346-023-01389-6.),
Answer1: We would like to thank the reviewer for this suggestion. We have added now the suggested review in the cited literature (Ref. 16).
- figure titles should be shorter and the descriptions contained in them should be moved to the text,
Answer 2: Thank you for your assessment. However, we think that the legend is important in order to understand the illustration. Shortening it would only be feasible if important information were lost.
- in lines 39, 49 and 78 there are unnecessary dashes in words,
Answer 3: We thank the reviewer for this hint and apologize for the incorrect dashes. We have checked again and removed the dashes.
- a summary of the results of cell studies in the form of tables would allow for better visibility of the results obtained.
Answer 3:Thank you for this advice. We have conducted a table with the main results of each cell line. This can be found at the end of the results section.
Reviewer 4 Report
Comments and Suggestions for Authors
This is an interesting experimental in vitro research showing the combined effects (cell morphology, toxicity, transcription factors expression) of Nimodipine and Vincristine on both cancer cell lines (A549, SAS and LN229) and neuronal cells (RN33B, SW10). The manuscript is clear, the methodology seems appropriate, and the results are well presented and discussed. I have only a few minor suggestion to improve the quality of the paper as follow:
The sentence at line 37 is not clear what it refers to, please explain.
I have some doubts about the necessity and usefulness of Figure 1.
Table S1 would be more effective if it reported the standard deviations of at least 3 experiments and the p-values ​​of a statistical test to demonstrate the significance of the differences observed.
I wonder if it would be useful to report, perhaps in the supplementary materials, the quantification of the bands normalized to the GAPDH control, of the other transcription factors not shown in Figures 7 and 9.
Author Response
This is an interesting experimental in vitro research showing the combined effects (cell morphology, toxicity, transcription factors expression) of Nimodipine and Vincristine on both cancer cell lines (A549, SAS and LN229) and neuronal cells (RN33B, SW10). The manuscript is clear, the methodology seems appropriate, and the results are well presented and discussed. I have only a few minor suggestion to improve the quality of the paper as follow:
The sentence at line 37 is not clear what it refers to, please explain.
Answer 1: Thank you very much for pointing this out! We have reworded the sentence and apologize for the inaccuracy.
I have some doubts about the necessity and usefulness of Figure 1.
Answer 2: Thank you for your assessment. It is our intention that the scheme will facilitate a more rapid comprehension of the study by the reader, including the details pertaining to the cell lines and experiments.
Table S1 would be more effective if it reported the standard deviations of at least 3 experiments and the p-values ​​of a statistical test to demonstrate the significance of the differences observed.
Answer 3: Thank you for this comment. We agree that further experiments with SD and p-values would be more significant. However, the table is intended to provide additional data to support the existing results, and thus we have analyzed one representative image from each cell line. The LDH assays are considered to be more meaningful, and thus this table has been included as a supplementary table.
I wonder if it would be useful to report, perhaps in the supplementary materials, the quantification of the bands normalized to the GAPDH control, of the other transcription factors not shown in Figures 7 and 9.
Answer 4: Thank you for pointing this out. We have added the requested quantifications into the supplementary material (Figure S1 and S2).